# Evaluation of Methane Emission Reduction Potential of Water Management and Chinese Milk Vetch Planting in Hunan Paddy Rice Fields

**Zhiwei Zhang [1,2], Jianling Fan [2], Yunfan Wan [1], Jinming Wang [1], Yulin Liao [3], Yanhong Lu [3] and Xiaobo Qin [1,\*]**

1   Institute of Environment and Sustainable Development in Agriculture, Chinese Academy of Agricultural Sciences, Ministry of Agriculture and Rural Affairs, Beijing 100081, China; zzw19961210@163.com (Z.Z.); wanyunfan@ami.ac.cn (Y.W.); wangjinming0305@163.com (J.W.)
2   School of Environmental Science and Engineering, Nanjing University of Information Science and Technology, Nanjing 210044, China; jlfan@nuist.edu.cn
3   Soils and Fertilizer Institute of Hunan Province, Changsha 410125, China; ylliao2006@126.com (Y.L.); luyanhong6376432@163.com (Y.L.)
\*   Correspondence: qinxiaobo@caas.cn

**Abstract:** In order to explore the methane reduction potential with two scenarios of water management and Chinese Milk Vetch return, we calculated the methane emissions of Hunan Province rice fields in 2019 using the SECTOR tool based on Excel and released by the International Rice Research Institute. Thus, we preliminarily established an agricultural carbon emissions monitoring, reporting, and verification (MRV) system. The results showed that: (1) There was significant spatial variation in methane emissions in Hunan rice fields, with higher emissions in both the south and north and lower emissions in the east and west. Late rice was the main contributor to methane emissions, and the cities of Changde, Hengyang, Yueyang, and Shaoyang were high-emission areas due to differences in rice planting types and areas. Compared with flooding (1275.75 Gg), optimized water management measures (mid-drainage and AWD irrigation) reduced methane emissions by 29~45% (905.79 and 701.66 Gg, respectively). (2) Under the same nitrogen input conditions, compared with a solely straw return (375.24 Gg), combining green manure with straw return could partially reduce methane emissions from Hunan super hybrid rice (327.63 Gg). Compared with the control fertilizers (404.28 Gg), the reduction rates of winter-planted Chinese Milk Vetch, the return of rice straw, and the incorporation of both Chinese Milk Vetch and straw were 7.19%, 13.01%, and 18.96%, respectively. Based on scientific accounting tools, a preliminary MRV system for rice field carbon emissions was established. Under the national demand for reducing fertilizer use and increasing efficiency, equal nitrogen organic amendments could effectively contribute to the development of green, low-carbon, and high-quality agriculture.

**Keywords:** rice fields; methane; water management; green manure; mitigation potential assessment

## 1. Introduction

Methane is a greenhouse gas in the atmosphere and is second only to carbon dioxide ($CO_2$), with a 100-year global warming potential that is 27.9 times that of $CO_2$ [1]. The Intergovernmental Panel on Climate Change (IPCC) reported that the atmospheric concentration of greenhouse gases has increased to a level unprecedented over the last 800,000 years, especially since 1750, when the methane concentration increased dramatically (150%) [2]. Human activities are the primary cause of the increased methane concentration in the atmosphere [3]. Rice fields are an important source of methane emissions in agriculture, accounting for 30% of global methane emissions [4]. In 2014, the methane emissions in agricultural activities in China reached 22,245 Gg, among which methane emissions from rice fields contributed 40.1% [5]. In June 2022, the Implementation Plan for Emission Reduction

and Carbon Sequestration in Agriculture in Rural Areas [6] was released, which clearly put forward important actions such as a reduction in methane emissions from rice fields. It highlighted a reduction in emission intensity per product unit through strengthening water and fertilizer application management, promoting the return of thoroughly decomposed organic fertilizers to the field, and breeding low-emission, high-yielding varieties. Hunan Province is the largest province of rice cultivation in China. Therefore, the study of technical measures and the potential of methane emission reductions in rice fields in Hunan is of great significance to the development of agricultural emission reduction measures in China.

Scholars in China have conducted a lot of research on methane emissions reduction technology in rice fields, which mainly focuses on water management and the optimization of organic material inputs [7–11]. The use of green manure and the return of straw to fields are effective measures that can increase yield and preserve fertility [12,13]. In particular, green manure is cultivated on unplanted land in the winter on a large scale in double-cropping rice planting areas in the southern region. It can improve the soil utilization rate and guarantee the nutrients required for rice production via its higher nitrogen-fixing capacity [14]. However, both green manure and straw contain much organic matter. Their return to fields can increase methane emissions in rice growing seasons [11,15,16]. With the state policies for reducing the use of fertilizers and pesticides, the measures of replacing chemical fertilizers with an equal number of organic fertilizers need to be studied urgently, and different combinations of green manure and straw when returned to fields also need to be optimized [17]. In particular, the effects of emissions reduction through saving fertilizers with an equal nitrogen input during the planting of Chinese Milk Vetch in the winter still need to be studied in depth [18]. Water management has a direct impact on methane emissions from rice fields. Mid-season drainage and Alternate Wetting and Drying (AWD) have been proven to have greater methane emissions reduction potential [7,19,20]. It can be seen that the development of rice field emissions reduction measures requires both the input of an organic amendment to improve the soil organic matter and soil productivity and the improvement of water management to reduce methane emissions. However, at present, there are still a few studies exploring the integrated effects of the return of an organic amendment to fields and water management on methane emissions [18].

To promote the development of low-carbon agriculture, a complete, scientific, and bottom-up monitoring, reporting, and verification (MRV) system needs to be established to evaluate the effects and potential of emission reduction measures. China has already established a sound MRV system [21–23]. Unlike industrial sectors, emissions in the breeding industry are non-point emissions; therefore, it is more difficult for accurate quantitative monitoring [24]. The study of the MRV system is still in the beginning stage, and relevant methodological studies need to be carried out to make improvements gradually at three levels, i.e., inventories, policies, and enterprises [24,25]. The estimation of emissions is the basis and key for obtaining a clear picture of the base number of agricultural carbon emissions and developing feasible emission reduction measures and an MRV system. The estimation methods for methane emissions from rice fields include the extrapolation method [26], the emission factor method [27,28], and the modeling simulation method [29–31]. The extrapolation method could be used to assess methane emissions at regional or larger scales based on the test data at a typical point. Wassmann et al. [26,27] estimated that, based on the point position data from rice fields in China, the total global methane emissions from rice fields in 1988 were $100 \pm 50$ Tg. However, methane emissions were affected by various factors such as soil characteristics [29], water conditions [7,32,33], and organic amendments [9,34]. The evaluation of global methane emissions based on point position data could cause great errors. In 2006, the IPCC published the Guidelines for National Greenhouse Gas Inventories [35]. It provided a three-tier method for estimating methane emissions from rice fields, with the main difference consisting of a selection of emission factors. The tier 1 method adopts default values recommended by IPCC. The tier 2 method is based on the specific emission factors and scale factors in different countries. The tier 3 method is a modeling method. Different countries can choose the appropriate method

according to the availability of activity data. There are many tools available for estimating greenhouse gas emissions (GHG emissions) from agriculture [36,37]. All these tools are based on the methodological framework of IPCC, and there are still fewer tools that are especially used for estimating GHG emissions from rice fields.

In this study, the Source-Selective and Emission-adjusted Greenhouse Calculator (SECTOR) for estimating GHG emissions from rice fields, published by the International Rice Research Institute (IRRI) and the geographic information system (GIS), was used to estimate the current status of methane emissions from rice fields and the emissions reduction potential at the municipal and provincial levels in Hunan province. This paper aimed to provide typical cases and basic support for the development of regional emission reduction policies and an agricultural MRV system by estimating the methane emissions from rice fields and the emission reduction potential in Hunan in 2019 under two technical measures for emissions reduction, i.e., optimizing water management and the planting of Chinese Milk Vetch in winter with equal nitrogen inputs, and evaluating the fertilizer and emission reduction effects of these two measures.

## 2. Materials and Methods

### 2.1. Introduction to the Research Area

Hunan province lies in the zone of transition from the Yunnan–Guizhou Plateau to the hilly areas in the south of the Yangtze River and from the Nanling Mountains to the Jianghan Plain in central China. It is between $24°38'$ and $30°08'$ in its north latitude and $108°47'$ and $114°15'$ in its east longitude, with a total area of 21,180,000 km$^2$. Hunan province is a typical producing area of double-cropping rice with a subtropical monsoon climate. It has abundant rainfall and ample sunlight hours throughout the year. Its average annual temperature ranges between 16.9 °C and 19.5 °C, and the average precipitation can reach 1159.8 to 2106 mm. In 2019, the rice planting area was 3,855,200 hm$^2$, and the total rice harvest was 26,115 Gg in Hunan province, accounting for 12.98% and 12.46% [38] of the national rice planting area and yield, respectively, making it the largest rice-producing province in China.

### 2.2. Research Method

#### 2.2.1. Sources of Rice Production Data

Data about the rice (single-cropping rice, early rice, and late rice) planting area, production per unit area, and the total production in Hunan province (provincial and municipal levels) in 2019 was obtained from the China Statistical Yearbook, Hunan Statistical Yearbook, and the statistical yearbooks of various cities in Hunan, respectively. The details are presented in Tables S1–S3 (please see Supplementary Information).

#### 2.2.2. Calculation Method of Methane Emissions from Rice Fields

The Tier 2 method provided in the Guidelines for National Greenhouse Gas Inventories [35] published by IPCC in 2006 was adopted in this paper, with the specific calculation method as follows:

$$CH_{4rice} = \sum_{i,j,k} \left( EF_{i,j,k} \cdot t_{i,j,k} \cdot A_{i,j,k} \cdot 10^{-3} \right) \tag{1}$$

where $CH_{4rice}$ is the annual methane emission, Gg·yr$^{-1}$; EF is the daily emissions factor of methane, kg CH$_4$ ha$^{-1}$ day$^{-1}$; t is the number of days of rice growth, day; A is the rice planting area, ha yr$^{-1}$; and i, j, k represents different ecosystems, water regimes, the number of organic amendments, and other conditions that can affect CH$_4$ emissions from rice fields, respectively.

Considering the changes in rice field management measures and the factors affecting methane production and emissions in rice fields, the daily emission factors of methane need to be adjusted according to the corresponding conditions as follows:

$$EF_i = EF_c \cdot SF_w \cdot SF_p \cdot SF_o \cdot SF_{s,r} \tag{2}$$

where $EF_i$ is the adjusted daily emissions factor for a specific rice harvest area; $EF_c$ is the baseline emissions factor for continuously flooded rice fields without organic additives; $SF_w$ is the field water management measures during the rice growth period; $SF_p$ is the water management measures before the rice planting period; $SF_o$ is the conversion factor for the change in the type and quantity of organic additives; $SF_{s,r}$ is the conversion factor for the soil and rice type.

The calculation tool was SECTOR (Version 2): a Microsoft Excel-based tool released by IRRI in 2021 (https://ghgmitigation.irri.org/knowledge-products/mrv-toolbox/sector (accessed on 2 April 2023)). Its operation procedure is shown in Figure 1, and the relevant calculation parameters are listed in Table 1.

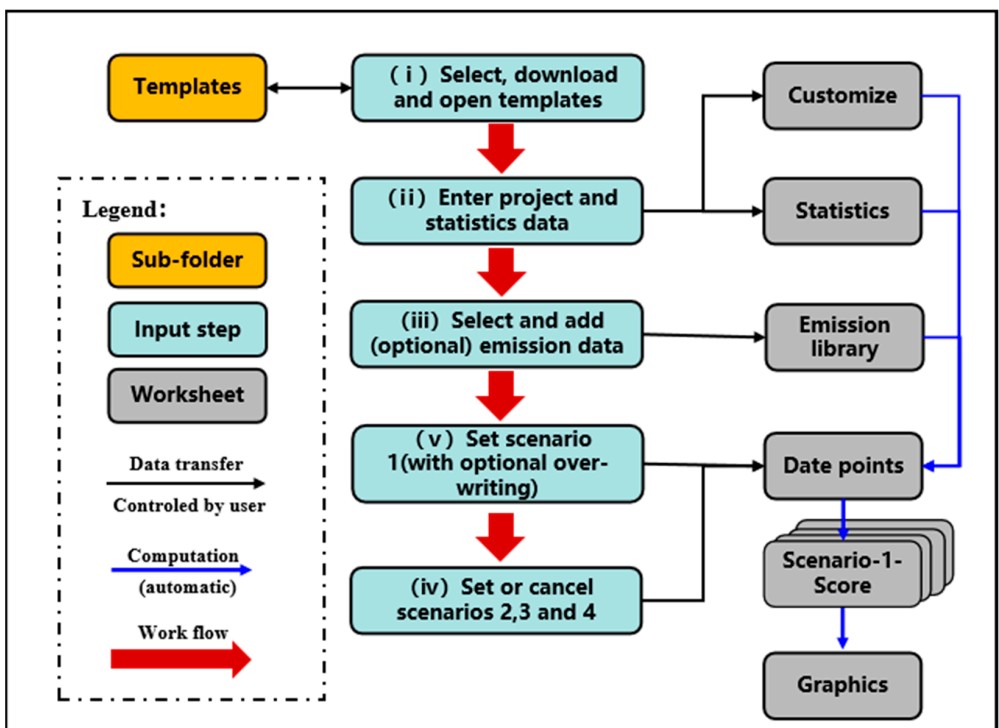

**Figure 1.** Flowchart of operations in SECTOR (Wassmann et al., 2019 [39]).

2.2.3. Chinese Milk Vetch Experiments Method

*(1) Site description and experimental design.*

The field experiment was conducted in 2021 at the Gaoqiao experimental base of the Hunan Academy of Agricultural Sciences in Hunan Province, China. All treatments were laid out in a randomized block design with three replicates each. Each plot measured 30 m$^2$ (4.0 m × 7.5 m). The four fertilization treatments were designed as follows: Chemical Fertilizer (CF), Straw full return+ chemical fertilizer (S), green manure return+ chemical fertilizer (M), and the Combined return of straws and green manure+ fertilizer (MS). The Chinese Milk Vetch (*Astragalus sinicus* L.) was grown in the winter and harvested and weighed as green manure before transplanting the single-season rice. Harvested green manure was then directly returned to the field with an average of 22.5 t ha$^{-1}$ applied to the three repeated experiments. The total amount of nutrients N, P, and K was the same between treatments, with N at 210 kg ha$^{-1}$, P$_2$O$_5$ at 75 kg ha$^{-1}$, and K$_2$O at 120 kg ha$^{-1}$. The N, P, and K content of Chinese Milk Vetch and straw were calculated before returning to the field, and part of the lacking was supplemented with chemical fertilizers. Water management included intermittent irrigation throughout the experiment except for mid-season drainage. Other field management measures were carried out according to local conventional methods. The basic soil properties in this experiment field and the nutrient contents of the tested Chinese Milk Vetch and straw are provided in Tables 2–4 [18].

**Table 1.** SECTOR model parameters used when comparing methane emissions for four rice fertilization scenarios in Hunan Province, China.

| Parameters | Description | Value | Source |
|---|---|---|---|
| Duration | The whole time of rice growth stage, day | Single-season rice: 120 [b], 110 [c] <br> Early rice: 82 [b] <br> Late rice: 131 [b] | Zheng et al., 2015 [40] |
| Yield | Calculated by statistic, t·ha$^{-1}$ | - | Yearbook |
| $EF_c$ | Daily emissions factor, kg $CH_4$·ha$^1$·day$^{-1}$ | Default value: 1.19 [a] <br> Early rice: 1.73 [b] <br> Single-season rice: 1.73 [b] <br> Late rice: 3.41 [b] | IPCC <br> Fu et al., 2010 [41] |
| $SF_p$ | Single-season rice/Early rice: non-flooded > 180 d; Late rice: non-flooded < 30 d | Early rice: 0.89 [a] <br> Single-season rice: 0.89 [a], 1 [c] <br> Late rice: 1 [a] | IPCC |
| $SF_o$ | The time of straw incorporation | Early rice/Single-season rice: 0.19 [a] <br> Late rice: 1 [a] | IPCC |
| Straw residue | t stubble·ha$^{-1}$·season$^{-1}$ | 0.8 [a], 0 [c] | - |
| Straw incorporation | Calculated by the ratio of straw amount and yield, t straw·ha$^{-1}$·season$^{-1}$ | 1:1 [b], 0 [c] | Gao et al., 2009 [42] |
| $SF_w$ | Water management with rice growth stage | Continuously flooded: 1 [a,c] <br> Mid-season drainage: 0.71 [a], 1 [c] <br> AWD: 0.55 [a] | IPCC |
| Straw | Only straw incorporation | - | - |

[a], indicates recommended value of IPCC; [b], comes from statistical yearbook and the literature; [c] was used for experimental treatments.

**Table 2.** Basic soil properties in this experimental field [18].

| Treatment | Organic Matter (g kg$^{-1}$) | Total N (g kg$^{-1}$) | Total C (g kg$^{-1}$) | C/N | DOC (mg kg$^{-1}$) | TDN (mg kg$^{-1}$) | Moisture (%) |
|---|---|---|---|---|---|---|---|
| Initial | 30.27 ± 1.02 | 1.98 ± 0.06 | 17.67 ± 0.59 | 8.90 | 190.00 ± 13.91 | 36.90 ± 4.25 | 26.73 ± 0.51 |
| CF | 29.70 ± 2.73 | 2.39 ± 0.42 | 29.87 ± 2.72 | 12.51 | 93.03 ± 9.65 | 32.87 ± 4.27 | 33.02 ± 0.66 |
| S | 27.63 ± 1.87 | 1.83 ± 0.08 | 27.80 ± 1.92 | 15.19 | 103.77 ± 4.78 | 39.10 ± 3.03 | 35.12 ± 0.08 |
| M | 29.73 ± 1.86 | 1.86 ± 0.05 | 29.90 ± 1.86 | 16.05 | 76.80 ± 3.71 | 31.30 ± 3.28 | 32.54 ± 0.27 |
| MS | 29.80 ± 2.21 | 1.94 ± 0.15 | 29.97 ± 2.24 | 15.45 | 78.00 ± 4.10 | 37.73 ± 2.89 | 31.66 ± 0.41 |

Chemical Fertilizer (CF), Straw full return + chemical fertilizer (S), green manure return + chemical fertilizer (M) and Combined return of straws and green manure + fertilizer (MS).

**Table 3.** Nutrient contents of tested Chinese Milk Vetch and straw [18].

| Item | Total N (g 100 g$^{-1}$) | Total P (g 100 g$^{-1}$) | Total K (g 100 g$^{-1}$) | Moisture (%) | Total C (g 100 g$^{-1}$) | Plant C/N |
|---|---|---|---|---|---|---|
| Chinese Milk Vetch | 2.46 | 0.176 | 1.29 | 89.7 | 39.4 | 16.02 |
| Straw | 1.24 | 0.264 | 2.44 | 50.9 | 37.5 | 30.24 |

**Table 4.** Nutrient and slaked lime input amount in different treatments (kg hm$^{-2}$) [18].

| Treatment | Chemical Fertilizer | | | Chinese Milk Vetch | | | Rice Straw | | | Total | | |
|---|---|---|---|---|---|---|---|---|---|---|---|---|
| | N | P$_2$O$_5$ | K$_2$O | N | P$_2$O$_5$ | K$_2$O | N | P$_2$O$_5$ | K$_2$O | N | P$_2$O$_5$ | K$_2$O |
| CF | 210 | 75.0 | 120 | | | | | | | 210 | 75 | 120 |
| S | 177 | 59.1 | 42.8 | | | | 32.6 | 15.7 | 77.2 | 210 | 75 | 120 |
| M | 153 | 65.7 | 84.0 | 57.0 | 9.33 | 36.0 | | | | 210 | 75 | 120 |
| MS | 120 | 49.8 | 6.81 | 57.0 | 9.33 | 36.0 | 32.6 | 15.7 | 77.2 | 210 | 75 | 120 |

*(2) CH$_4$ emissions and flux measurements.*

Methane flux was measured once every three days or once per week for later mid-drainage throughout single-season rice with static closed chambers. The sample box was composed of a base and box body with a height of 100 cm and a length and width of 37 cm. The box material was made of PVC, and the side wall was covered with a reflective heat insulation film to prevent the temperature inside the box from changing too much during sampling. The CH$_4$ flux was determined throughout each rice growing period using the closed chamber gas chromatograph (GC) method. A hydrogen flame ionization detector (FID) was installed in gas chromatography, and the FID linear response range met the requirements. A standard gas sample with a CH$_4$ concentration of 1.69 ppm was tested 6 times in FID, and the standard deviation was about $\pm0.012$ ppm; the coefficient of variation CV = 0.64% and was also less than 1%. The standard gas used in the measurement process was provided by the Standard Gas Research Center of the National Metrology Institute: the concentration of CH$_4$ standard gas was 1.69 ppm. The detector temperature was 200 °C, the column type was Porapack Q, the column temperature was 70 °C, the carrier gas was N$_2$ (>99.999%), and the carrier gas flow rate was 25 mL min$^{-1}$. The daily average flux and standard error for the CH$_4$ were calculated from triplicate plots.

$$F_c = k \cdot V / A \cdot dC / dt \cdot \rho \cdot 273 / (273 + T) \cdot P / 1013 \tag{3}$$

where $F_c$ is the flux of trace gas (CH$_4$: mg m$^{-2}$ h$^{-1}$); $dC$ is the difference between the initial and final GHG concentrations ($10^{-6}$ mol mol$^{-1}$) during the duration of closure ($dt$: in h); $V$ is the headspace volume of the chamber (Formula (3)); A is the bottom area of the chamber (m$^2$); $\rho$ is the density of GHG at 273 K and 1013 hPa (0.717 g L$^{-1}$ for CH$_4$); $T$ is the mean air temperature in the chamber during closure (°C); $P$ is the air pressure during incubation (hPa), which was taken to be ~1013 hPa because the air pressure tube was installed in the chamber to maintain the balance of pressure inside and outside the chamber during sampling; and $k$ is a coefficient for dimensional conversion.

2.2.4. Setting of Scenarios for Methane Emissions from Rice Fields

*(1) Water management measures.*

Three water management regimes, namely flood irrigation, mid-season drainage, and AWD irrigation, were set according to the rice field water management characteristics. In SECTOR, different scale factors (SF) were set according to the water management measures before and during the growing season and different rice straw treatment methods. The SF$_p$ for pre-season flooding management was determined as 1 for late rice and 0.89 for early rice and single-cropping rice, mainly based on the consideration that the pre-season water management (in the idle period in winter or between early and late rice planting periods) needed no flooding or the flooding time was less than or equal to 30 days. For water management during the growing season, SF$_w$ was divided into three, namely, flooding, mid-season drainage, and Alternate Wetting and Drying (AWD), with the values of 1, 0.71, and 0.55, respectively. Straw was not burnt but was returned to fields immediately after harvesting in the season. SF$_o$ was determined to be 0.19.

Two scenarios, i.e., the IPCC recommended value and regional value, were set for the selection of daily emission factors of methane from rice fields (both scenarios were also used for the assessment of the spatial variability characteristics of methane emissions

at a municipal level and the potential of emission reductions). We adopted the IPCC recommended value based on the daily emissions factor of 1.32 kg·$CH_4$·$ha^{-1}$·$day^{-1}$ in Asia. This value was lower than the daily emission factors of methane from rice fields in China and did not distinguish between planting types. The research results of Fu et al. [41] were selected as the regional value of the emissions factor for comparison. The selection of daily methane emission factors for each scenario is detailed in Table 1.

We also set up two scenarios, i.e., the return of straw to fields and no return of straw to the fields, for a comparison based on the different water management scenarios. The default value of 0.8 t·stubble·$ha^{-1}$·$season^{-1}$ was used for stubbles. The number of straw returned to the fields was calculated based on the dry matter partitioning (1:1) to seed and stems.

*(2) Return of different organic amendment to fields with equal nitrogen inputs.*

Methane seasonal emissions were obtained with different treatments from rice fields [18]. The duration was 110 d for single-season rice. The daily emission factors between these treatments were calculated and followed as: 2.4873 kg·$CH_4$·$ha^{-1}$·$day^{-1}$ for CF, 2.1637 kg·$CH_4$·$ha^{-1}$·$day^{-1}$ for S, 2.3086 kg·$CH_4$·$ha^{-1}$·$day^{-1}$ for M, and 2.0157 kg·$CH_4$·$ha^{-1}$·$day^{-1}$ for MS. The details are in the supplementary materials (please see Supplementary Information) and Table 1. Contrary to the general situation, field experiment results showed that CF treatment had higher methane emissions. A possible reason for this is that an organic amendment such as straw and Chinses Milk Vetch undergoes a fully aerobic decomposition because of its return for a long time. The promotional effect of methane emission was, therefore, reduced.

### 2.3. Data Processing and Plotting

The mean and standard error of the experimental methane emission data with different treatments were collated and preliminarily calculated using Microsoft Excel (v. Office 365, Microsoft). The degrees of freedom were calculated using the Satterthwaite method. The means were separated using Fisher's protected least significant difference test at the 0.05 significance level, and the Tukey–Kramer method was used for *p*-value adjustment at a significance level of 0.05 [43]. Using Origin (v. 2023) to map the methane emissions of yield. Spatial distribution maps of methane emissions were drawn using Arc GIS 10.5 (ESRI, Environmental Systems Research Institute, Redlands, CA, USA).

## 3. Results

### 3.1. Spatial Variability Characteristics of Methane Emissions from Rice Fields in Different Cities in Hunan

In 2019, the variability of methane emissions from rice fields in different cities in Hunan province (except Zhangjiajie, for which statistical yearbook data were not collected) was large, appearing high in the southern and northern regions and low in the eastern and western regions as a whole (Figure 2a–d). When IPCC-recommended values were adopted, the methane emissions from rice fields in municipal administrative units were Changde > Hengyang > Yueyang > Shaoyang > Yongzhou > Yiyang > Changsha > Chenzhou > Zhuzhou > Huaihua > Loudi > Xiangtan > Xiangxi > Zhangjiajie. Among these, Changde, Hengyang, Yueyang, and Shaoyang were high methane emission areas, with a total methane emission of 277.66 Gg, accounting for 46.09% of the total emissions in the province. The methane emissions in Zhangjiajie, Loudi, Xiangtan, and Xiangxi were relatively low, with a total of 73.63 Gg, accounting for only 12.22% of the total emissions in the province (Figure 2a). When the regional values of daily methane emission factors were adopted, the order of methane emissions in different cities changed slightly as Hengyang > Changde > Yueyang > Shaoyang > Yongzhou > Yiyang > Changsha > Chenzhou > Zhuzhou > Loudi > Xiangtan > Huaihua > Xiangxi > Zhangjiajie. Changde, Hengyang, Yueyang, and Shaoyang remained the four areas with high methane emissions (609.52 Gg), and the proportion of the total methane emissions in these four cities increased slightly to 47.78%.

In Zhangjiajie, Xiangtan, Huaihua, and Xiangxi, where the emissions were relatively low, the total emissions were reduced to 129.73 Gg, accounting for 10.17% only (Figure 2b).

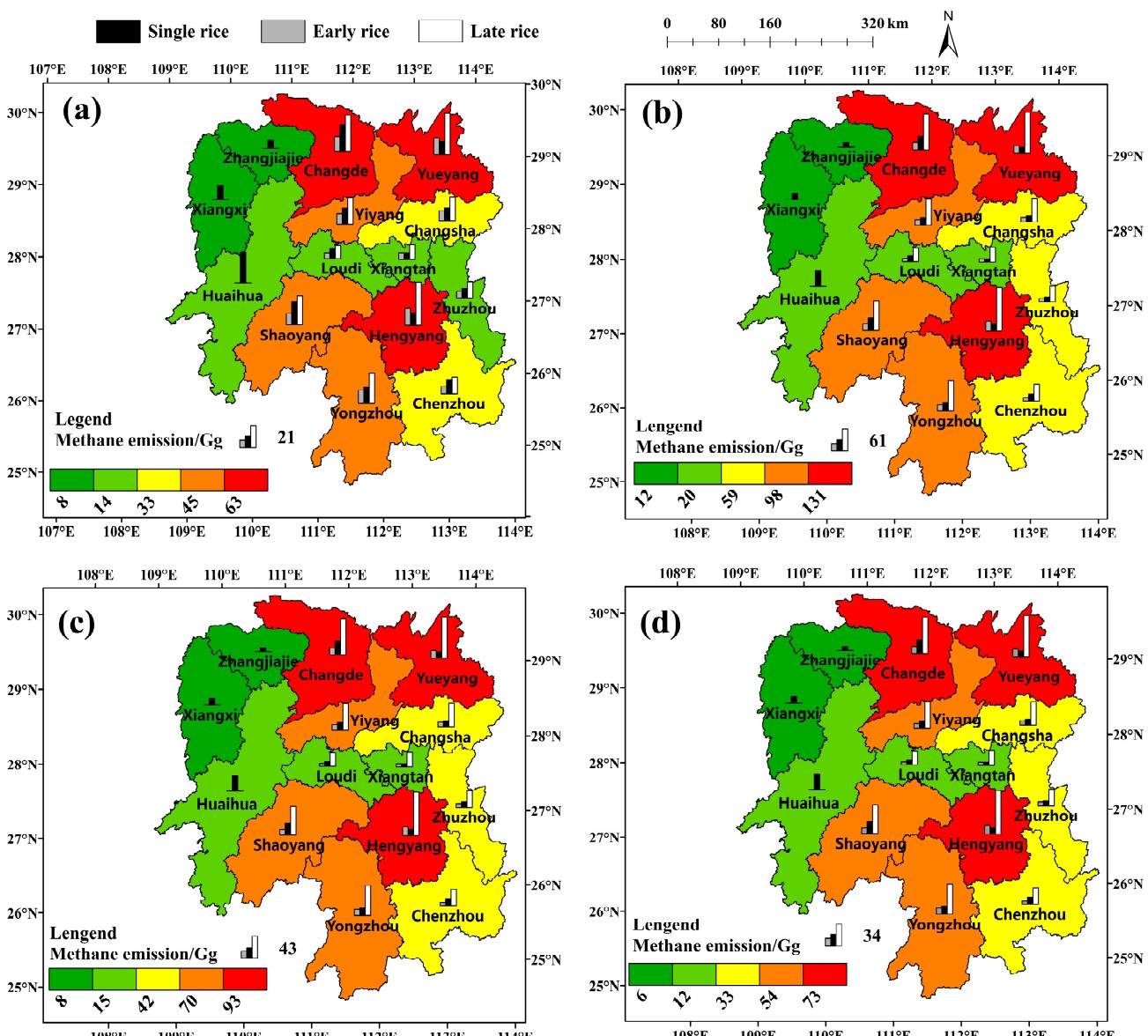

**Figure 2.** Methane emissions from rice fields in different cities in Hunan in 2019: (**a**) IPCC recommended value-methane emissions from flood irrigation; (**b**) Regional value-methane emissions from flood irrigation; (**c**) Regional value-methane emissions from mid-season drainage; (**d**) Regional value-methane emissions from AWD irrigation.

The water management of mid-drainage and AWD irrigation had common characteristics (Figure 2c,d). In terms of the planting type, methane emissions in the late rice season contributed the greatest part. In the four cities with a high methane emission value, i.e., Changde, Hengyang, Shaoyang, and Yiyang, the implementation of mid-season drainage and AWD irrigation was considered in the late rice season. Their emissions reduction potential could reach 176.76 Gg and 274.28 Gg, respectively (Figure 2b–d).

### 3.2. Assessment of the Emission Reduction Potential of Water Management

With flood irrigation, different daily methane emission factors resulted in big differences in methane emissions. When calculated with the emission factors recommended by IPCC, the methane emissions from rice fields in Hunan were 602.41 Gg, with the highest emission from late rice (283.32 Gg), followed by that of single-cropping rice (204.15 Gg) and the lowest emissions were from early rice (114.95 Gg). When calculated based on their regional value, the total emissions were 1275.75 Gg, which was 2.12 times the former.

Water management is one of the important factors affecting methane emissions from rice fields. In China, mid-season drainage has been widely used as a conventional field management method. Mid-season drainage could not only reduce ineffective tillering and improve rice yield [44] but also significantly reduce methane emissions from rice fields [45]. In this study, the emission reduction potential of mid-season drainage and AWD irrigation was compared with flood irrigation (using regional daily methane emission factors) as the baseline emission. The methane emissions from rice fields in the whole province after one mid-season drainage were 905.79 Gg, with an emissions reduction potential of 369.97 Gg (at a reduced rate of 29%). When the AWD carbon reduction technology was adopted in the province, the methane emissions from rice fields were reduced to 701.66 Gg, with an emissions reduction potential of up to 574.09 Gg and an emissions reduction rate of 45%.

The return of straws to fields is a more economical and effective measure for carbon sequestration in rice fields [44]. Under the guidance of national policies, the rate of return has increased over the years, but the direct return of straw to fields could also bring an external organic amendment, thereby increasing the content of substrates for forming methane. In addition, straw degradation also consumes oxygen in the soil and reduces Eh, thus creating favorable conditions for methanogens and resulting in increased methane emissions [46]. In this study, the methane emissions and emission reduction potential of flood irrigation, mid-season drainage, and AWD irrigation were compared and analyzed based on the return of straw to fields. The results showed that the methane emissions from rice fields with flood irrigation were 2717.69 Gg. Compared with the fields with no return of straws, methane emissions increased by 1441.94 Gg at a rate of 113.03%. Considering water management emission reduction technology, the methane emissions from rice fields with mid-season drainage and AWD irrigation were 1929.56 and 1494.73 Gg, respectively, with methane emissions reduced by 788.13 and 1222.96 Gg, respectively, compared to flood irrigation. Considering the rice planting type, methane emissions from late rice were the largest, contributing 82.41% to methane emissions in the province (Table 5).

**Table 5.** Methane emissions and mitigation potential of rice paddies in Hunan Province in 2019 under different water management measures.

| Scenario | Type of Planting | | | | Mitigation Potential (Gg) |
|---|---|---|---|---|---|
| | Methane Total (Gg) | Single-Cropping Rice (Gg) | Early Rice (Gg) | Late Rice (Gg) | |
| Flooded irrigation [a] | 602.41 | 114.95 | 204.15 | 283.32 | - |
| Flooded irrigation [a] | 1275.75 | 167.11 | 296.78 | 811.86 | - |
| Mid-season drainage [b] | 905.79 | 118.65 | 210.72 | 576.42 | 369.97 |
| AWD irrigation [b] | 701.66 | 91.91 | 163.23 | 446.52 | 574.09 |
| Flooded irrigation [c] | 2717.69 | 251.28 | 477.93 | 1988.47 | - |
| Mid-season drainage [c] | 1929.56 | 178.41 | 339.33 | 1411.82 | 788.13 |
| AWD irrigation [c] | 1494.73 | 138.21 | 262.86 | 1093.66 | 1222.96 |

[a] and [b] represent different methane emission factors from rice paddies: [a] is the recommended value by IPCC; [b] is a regional value based on the regional value in Hunan Province; [c] means methane emissions with straw incorporation at different water management.

The methane emissions of yield were calculated under different water management and straw returning, and the results are shown in Figure 3. Different from the daily emission factor recommended by IPCC, China's rice fields were presented as having higher methane emissions. Without straw returning, methane emissions were 43.69 kg t$^{-1}$. Optimized water management (MD, AWD) could reduce methane emissions by 12.67, 19.66 kg $CH_4$ per ton of rice production. Straw returning significantly increased methane emissions from flooded paddy fields (93.19 kg t$^{-1}$). Additionally, it was also an effective method for improving rice yield. In China, the returning of straw is widely promoted. Therefore, MD and AWD technology could partially offset increased $CH_4$ emissions associated with straw amendments (27.03, 41.94 kg t$^{-1}$).

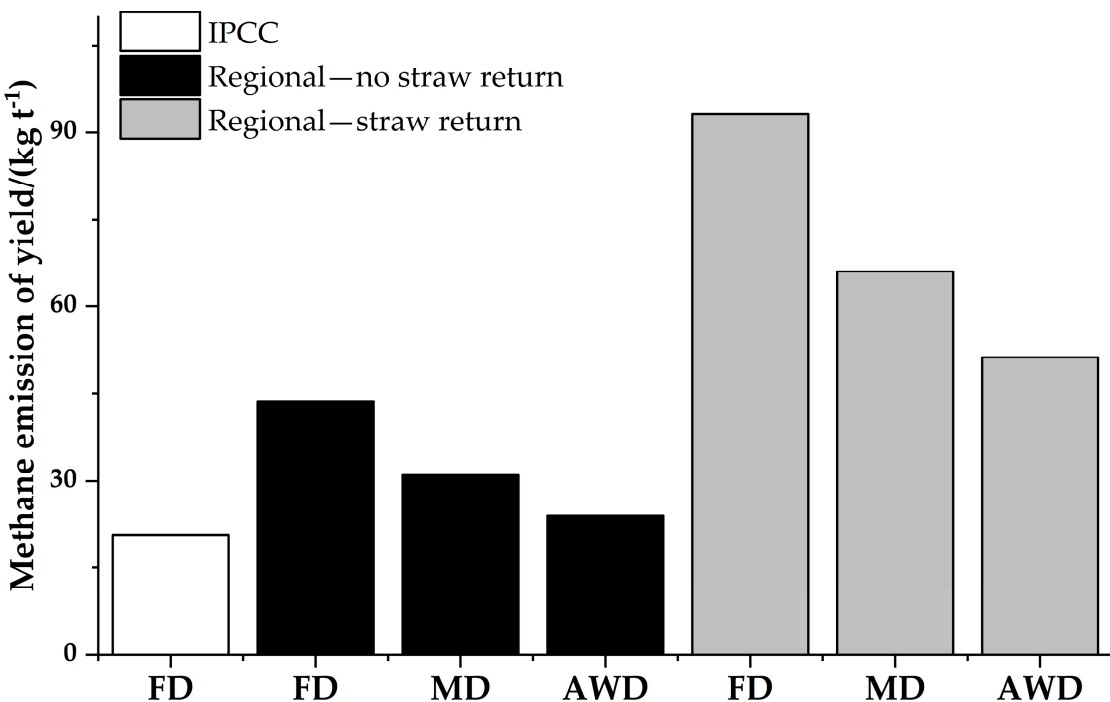

**Figure 3.** Methane emission of yield with different water management and straw return. (FD, Flooded irrigation; MD, Mid–season drainage; AWD, Alternate wetting and drying).

*3.3. Assessment of Methane Emission Reduction Potential of Winter-Planting of Chinese Milk Vetch under Equal Nitrogen Conditions*

3.3.1. Assessment of Methane Emission Reduction Potential

The methane emissions from indica hybrid rice with different treatment measures under equal nitrogen conditions in different cities in Hunan province in 2019 were arranged in order of CF (404.28, 11.07~57.79 Gg) > M (375.24, 10.27~53.64 Gg) > S (351.69, 9.63~50.27 Gg) > MS (327.63, 8.97~46.83 Gg) (Table 6). Among them, CF was the baseline treatment with a pure fertilizer application and the return of straw to fields without planting Chinese Milk Vetch, while S, M, and MS treatments were compared with the inputs of different organic amendments. Different types and methods of return in organic amendments to fields could lead to different effects of methane emissions reduction. Compared with CF, the reduction rates of S and M treatments were 13.01% and 7.19%, respectively. The emission reduction potential of the winter planting of Chinese Milk Vetch combined with the return of straw to fields (MS) was higher, reaching 18.96%. With the same condition of organic amendment return to fields, the emissions of MS were also 6.68% and 12.69% lower than that of S and M treatments, respectively. In Hunan, double-cropping rice is typical, but in recent years, the area of single-cropping rice has increased year by year, showing a trend of transformation from double-cropping rice to single-cropping rice [47]. Despite the higher yield of single-cropping super rice, its methane emissions also need attention. The return

of straw and green manure to fields is a conventional method for carbon sequestration in the soil, but methane emissions from super rice fields with equal nitrogen inputs still need further discussion. The results of the preliminary experiment performed by the task group showed that the emissions from S and M treatments combined with water management were slightly lower than those with the CF treatment. With CF treatment as the baseline scenario, S and M treatments had a higher emission reduction potential and could reduce the methane emissions from single-cropping rice by 52.60 (1.44~7.52) and 29.05 (0.80~4.15) Gg, respectively. The methane emission reduction of M treatment was 7.19%. It could reduce the amount of application in chemical nitrogen fertilizers by about 27.15% compared with the CF treatment and by 13.78% compared with the S treatment [18]. At a provincial level, the emissions from a combined return of green manure and straw to the fields (MS) were lower than that from the return of the two separately, and the methane emissions from S and M treatments were reduced by 24.06 and 47.61 Gg, respectively. In conclusion, under equal nitrogen conditions, methane emissions reduction received a better effect from the winter planting of Chinese Milk Vetch combined with the return of straw to the fields (MS). It could enhance soil nutrients and improve the pools of carbon and nitrogen [48] while reducing carbon emissions from rice fields as an effective measure for rice yield preservation and emissions reduction.

**Table 6.** Methane emissions and mitigation potential of single-season rice paddies under equal nitrogen conditions with winter-planted Chinese Milk Vetch and straw incorporation (Gg).

| Cities | Treatments | | | | Mitigation Potential | | |
|---|---|---|---|---|---|---|---|
| | CF | S | M | MS | S | M | MS |
| Xiangtan | 11.07 | 9.63 | 10.27 | 8.97 | 1.44 | 0.80 | 2.10 |
| Zhuzhou | 14.78 | 12.86 | 13.72 | 11.98 | 1.92 | 1.06 | 2.80 |
| Loudi | 18.41 | 16.01 | 17.09 | 14.92 | 2.39 | 1.32 | 3.49 |
| Changsha | 18.60 | 16.18 | 17.27 | 15.08 | 2.42 | 1.34 | 3.53 |
| Yueyang | 23.50 | 20.44 | 21.81 | 19.04 | 3.06 | 1.69 | 4.46 |
| Hengyang | 24.35 | 21.19 | 22.60 | 19.74 | 3.17 | 1.75 | 4.62 |
| Xiangxi | 24.75 | 21.53 | 22.97 | 20.06 | 3.22 | 1.78 | 4.69 |
| Chenzhou | 26.20 | 22.79 | 24.32 | 21.23 | 3.41 | 1.88 | 4.97 |
| Yiyang | 27.22 | 23.68 | 25.27 | 22.06 | 3.54 | 1.96 | 5.16 |
| Yongzhou | 30.24 | 26.30 | 28.07 | 24.51 | 3.93 | 2.17 | 5.73 |
| Shaoyang | 31.18 | 27.13 | 28.94 | 25.27 | 4.06 | 2.24 | 5.91 |
| Changde | 44.94 | 39.09 | 41.71 | 36.42 | 5.85 | 3.23 | 8.52 |
| Huaihua | 51.26 | 44.59 | 47.57 | 41.54 | 6.67 | 3.68 | 9.72 |
| Zhangjiajie | 57.79 | 50.27 | 53.64 | 46.83 | 7.52 | 4.15 | 10.96 |
| Total | 404.28 | 351.69 | 375.24 | 327.63 | 52.60 | 29.05 | 76.65 |

Calculation of $CH_4$ emission reduction potential using CF treatments as the baseline scenario. Chemical fertilizer (CF), straw full return+ chemical fertilizer (S), green manure return + chemical fertilizer (M) and combined return of straws and green manure + fertilizer (MS).

### 3.3.2. Influence of Organic Amendment Incorporation

In terms of rice yield, it was ranked from high to low among the different treatments as follows: M > S > MS > CF (Table 7). Returning an organic amendment to the fields could effectively increase the rice yield per unit area. Compared with CF, the treatments of S, M, and MS increased the rice yield by 2.40%, 2.94%, and 2.19%, respectively. Rice treated with M had the best yield increase. The yield of combined straw and Chinese Milk Vetch incorporation was lower. However, MS treatments had a better effect on reducing emissions.

**Table 7.** Influence of different fertilization treatments on rice yield.

| Treatment | | Yield/(kg ha$^{-1}$) |
|---|---|---|
| CF | I | 9116.67 |
| | II | 8500.00 |
| | III | 9066.67 |
| | Average | 8894.44 |
| S | I | 9566.67 |
| | II | 9108.33 |
| | III | 8650.00 |
| | Average | 9108.33 |
| M | I | 9433.33 |
| | II | 9133.33 |
| | III | 8900.00 |
| | Average | 9155.56 |
| MS | I | 9016.67 |
| | II | 9200.00 |
| | III | 9050.00 |
| | Average | 9088.89 |

I, II, III represent repetitions of different experimental treatments.

## 4. Discussion

### 4.1. Emission Reduction Potential of Water Management and Winter-Planting of Chinese Milk Vetch

For methane emissions from rice fields, water management is one of the most influential measures [49,50]. In addition to this, rice varieties, tillage, and fertilization also have important effects [51–54]. Methane can be produced in rice fields only when methanogens act on organic substrates in a strictly anaerobic flooded environment [55]. Water management emission reduction technologies (mid-drainage and AWD irrigation) can improve soil permeability by extending field aeration. The increased oxygen content can inhibit the activity of methanogens so that methane-oxidizing bacteria (methanotrophs) can oxidize methane more easily, thus achieving the emissions reduction effect [7,19]. In response to the need for emissions reduction and water conservation, the water management model in southern rice areas of China gradually changed from a traditional flood irrigation model to an intermittent irrigation model (flood irrigation, mid-season drainage) [56]. Generally, methane emissions from rice fields reach their peak at the early stage of rice growing, while the tillering stage is the period with high methane emissions [57]. In this period, rice is grown vigorously, and photosynthesis is strong. The increased root secretions lead to enhanced microbial activity, and, correspondingly, the decomposition of organic amendments, which produce enough substrates for methanogens [58]. Therefore, reasonable water management (AWD) at the tillering stage can not only reduce methane emissions from rice fields by 36–77% [41,59] but also reduce the ineffective tillering of rice roots and increase the yield. In this paper, methane emissions from rice fields after mid-drainage in Hunan province was 905.79 Gg, as calculated based on the mid-season drainage scale factor of IPCC. Compared with flood irrigation, this could reduce methane emissions by 369.97 Gg, with a reduction rate of 29%. In AWD irrigation, the cycles of alternate wetting and drying were increased on the basis of mid-season drainage. Its mechanism of emission reduction was consistent with that of mid-season drainage and could also stabilize or even increase yields [60,61]. This technology has been widely used in Southeast Asia [62]. Studies have evaluated the applicability of this technology in rice planting areas in Hunan [63]. This study found that if this technology was promoted in Hunan, the theoretical emission reduction potential could reach 574.09 Gg with an emission reduction rate of 45%.

Sustainable agricultural development requires the improvement of soil fertility, the reduction in fertilizer application, and the consideration of emissions reduction. Rice straws and green manure are common organic amendments in rice fields, and their single or combined return in different forms is considered an effective means of improving soil

productivity and enhancing soil carbon and nitrogen pools [14]. A lot of research has shown [7,15,64] that the return of straw to fields can promote methane emissions during the rice growing period by increasing external and usable carbon sources such as active organic matter in the soil. In this paper, it was estimated that under flooded conditions, the methane emissions from the return of all straws to fields in the Hunan province would be 2717.69 Gg. Compared with no return of straw to the fields, the rate of emissions increased to 113.03%. If mid-drainage and AWD irrigation technologies were implemented, the emissions reduction potential could exceed 700 Gg. This shows that the emission promotion effect of the return of straw to fields could be offset to some extent by water regulation. Green manure not only enhances the carbon and nitrogen content and fertility of the soil but also has a lower positive effect on methane emissions compared to the return of straw to fields [18]. The reason for this is that green manure has a lower C/N ratio and shorter decomposition time, and under the priming effect, the proliferation of microorganisms can consume soil organic matter, which reduces the DOC content [17] and the carbon sources usable by methanogens, thus resulting in lower methane emissions. By contrast, straw contains lignin and other macromolecules that are difficult to decompose; therefore, it takes a longer time for them to decompose. This may last for the whole rice growing period or longer, which can provide a stable carbon source for methanogens. Therefore, emission promotion effects can last for the whole rice growing season, resulting in higher emissions. The results of the 2021 field experiment showed that seasonal emissions under CF treatment were higher than that under S, M, and MS treatment. The possible reason for this is that, in this experiment, straw was returned to the field immediately after harvesting (October 2020), while the Chinese Milk Vetch was returned in the full-bloom stage (April 2021). Before transplanting, it took more than 6 months for the straw to return and about 2 months for Chinese Milk Vetch to return. Two organic amendments had enough time for aerobic decomposition. Therefore, active carbon sources decreased for methanogens during the rice growth period, and methane emissions were lower. However, the return of the organic amendment can promote the activity of soil microorganisms and simultaneously consume part of soil carbon sources while decomposing straw and Chinese Milk Vetch. Laboratory culture experiments based on field experiments also verified this result [18]. Compared with the CF treatment, the methane production potential of the M treatment was lower. This indicates that most of the organic amendment increased after returning to the field and being broken down by decomposition. Additionally, Chinses Milk Vetch combined with straw returning had a higher emission reduction potential. After straw incorporation, soil microbial activity increased with a large amount of soil carbon consumption. However, this process was activated again after returning Chinese Milk Vetch, which further reduced the available carbon source in the field. MS treatment had a higher methane emissions reduction (Table 6). The use of chemical fertilizers achieved zero growth in China in 2017, but there was still a significant nitrogen surplus in the number of fertilizers applied to farmland [65]. For rice, the nitrogen surplus was about 70 kg N ha$^{-1}$ [66]. Against the background of the state strategic objectives of carbon peaking and carbon neutrality and the need for high-quality development, the replacement of chemical fertilizers with organic fertilizers that have an equal amount of nitrogen is a potentially effective means for reducing surplus nitrogen in fields, stabilizing yields and increasing the content of organic matter in the soil [67]. The results of this study indicate that there is great potential for the return of green manure and straw with an equal amount of nitrogen in the main rice-producing areas of Hunan.

### 4.2. Estimation Method and Establishment of MRV System for Methane Emissions from Rice Fields

The rice growing areas in China span a wide range with different soil and climate conditions. The methane emissions from rice fields in different areas are also different. Studies have shown [68] that methane emissions from rice fields in inland areas are higher than those from coastal areas. Owing to such a great temporal and spatial variation [69], it is difficult to accurately estimate the emissions at regional and national levels. Internationally,

the main methods for estimating methane emissions from rice fields are now the emission factor method and the modeling method. The emission factor method is the assessment method officially recommended by IPCC as well as the method used by many countries for preparing the inventory. When the emission factor method is used for estimating methane emissions from rice fields, the selection of the emission factor may have a significant effect on the estimation results. In this study, the methane emissions from the rice planting areas of Hunan were estimated using the tool published by IRRI and based on the IPCC-recommended values and regional values, respectively. The results showed that methane emissions from rice fields based on the regional value was 2.12 times that based on the IPCC recommended value. Our estimation result was similar to the result estimated with the DNDC model by Wang et al. [29]. It can be seen that under the production conditions in China, the adoption of monitoring-based emission factors in China is needed to obtain more accurate estimates. However, estimates based on emission factors inevitably lead to a high correlation between emissions and the planting area. According to this study, under flooding conditions, the methane emissions from rice fields could take on the spatial characteristic of a high value in the southern and northern areas and a low value in the eastern and western areas of Hunan province. Changde, Yueyang, Hengyang, and Shaoyang were the areas with high methane emissions, and the cultivation area was a major cause of regional differences in emissions [69]. This was consistent with the change in rice planting areas in different cities (Figure 2a–d). In 2018, the Ministry of Ecology and Environment published its second two-year update report on climate change in the People's Republic of China [5]. It pointed out that in 2014, methane emissions from rice fields in China were 8911.0 Gg. According to the proportion of rice planting areas in Hunan province compared to that in the whole country, the emissions in Hunan were about 1158.43 Gg, which is close to the estimation in this study (Table 1). Related studies [5,27,70,71] showed that methane emissions from rice fields in China were estimated to range between 7410 and 8910 Gg according to the emission factor method. Estimated based on the proportion of rice planting areas in Hunan compared to that in the whole country [38] (the multi-year proportion of the planting area was at 13%), the methane emissions ranged between 960 and 1160 Gg, which is similar to the estimates of no straw return in this study but is also significantly lower than the methane emissions with the return of straw to the fields.

The emissions of methane from rice fields were a net result of the generation, re-oxidation, and transmission of methane [54,72,73]. Huang et al. [31] developed a semi-empirical model, CH4MOD, to simulate the methane emissions from flooded rice fields by establishing numerical relationships in various ecosystem processes, which was well-tested in rice field experiments in the US. Considering the effect of bubble transport, Huang et al. [74] further improved the model, thus making the model results more accurate. CH4MOD has been well-verified in China [75,76]. This model is also one of the models recommended by IPCC as Tier 3 of the Guidelines for National Greenhouse Gas Inventories. Its advantages are that the input parameters are few and easy to obtain, and it can be combined with GIS technology to subdivide the research area according to a certain spatial scheme to estimate the methane emissions from the research area and improve their accuracy [77]. With CH4MOD, Zhang et al. [78] estimated that the methane emissions from rice fields in China ranged between 4130 and 6850 Gg from 1960 to 2009. Sun et al. [79] estimated that the emissions in 2015 were 4750 (4190–5610) Gg. By 2014, CH4MOD was used to prepare the inventory of methane emissions from rice fields in China. A possible reason why the data value from the national inventory (8911.0 Gg in 2014) was higher than the data value from the above research is that there were differences in the selection of regional emission factors. The Microsoft Excel-based methane emission estimation tool (SECTOR) published by IRRI was selected for this study. Compared with CH4MOD, it was easy to use. SECTOR calculated the emission reduction potential of different emission reduction measures based on the IPCC Tier 2 method. For example, in this study, the methane emissions and reduction potential of water management and the return of organic amendments to fields were estimated using an easier method [41]. The entire methodologi-

cal framework complied with international treaties and could provide a reference for policy development and implementation. In fact, SECTOR is only one of many tools that IRRI has released to build MRV platforms. After this platform was improved, a complete MRV system for the applicability assessment, emission reduction estimation, and cost–benefit analysis of emissions reduction was constructed [39,80,81]. At present, the MRV system for agriculture in China is not perfect. In comparison with developed countries [82], the MRV system for agricultural carbon emissions in China is still at a preliminary stage, and there is an urgent need to promote the application of such tools and platforms. The development of estimation methods, activity level data, and emission factors that are applicable in regions has limited the construction of the MRV system. In particular, compared with the mature mechanism for reporting GHG emissions in developed countries, China still lags behind in the routine collection of activity level data. Data are collected mainly from local research projects, and there is a lack of a routine and systematic reporting mechanism. In addition, improvements need to be made with regard to laws and regulations, systemic building, enterprise specifications, and methodologies to establish a scientific, rational, and feasible MRV system for agriculture [25].

## 5. Conclusions

1. The potential of methane emission reductions in rice planting areas in Hunan was estimated based on the MRV tool published by IRRI. Compared with flood irrigation, the adoption of optimized water management measures (mid-season drainage and AWD irrigation) could result in a potential reduction rate of 29–45%. There is great spatial variability in methane emissions at the municipal level in rice planting areas in Hunan. Changde, Hengyang, Yueyang, and Shaoyang are four regions with high methane emissions. This is mainly caused by the differences in rice planting types and areas.

2. Under equal nitrogen conditions, compared with the return of green manure or straw alone to fields, the return of both green manure and straw to fields can partially inhibit methane emissions from indica hybrid rice in Hunan. Compared with the application of chemical fertilizers alone, the emission reduction rate of winter planting Chinese Milk Vetch, the return of straw to fields alone, and the return of straw with green manure could reach 7.19%, 13.01%, and 18.96%, respectively. With a national need to reduce their application and increase the effects of chemical fertilizers, the replacement of chemical fertilizers with the organic amendment of equal nitrogen content could effectively promote the green, low-carbon, and high-quality development of agriculture.

**Supplementary Materials:** The following supporting information can be downloaded at: https://www.mdpi.com/article/10.3390/agronomy13071799/s1, Table S1: The data of rice data in Hunan Province in 2019; Table S2: The data of rice sown area and yield in different cities in 2019; Table S3: The data of rice sown area and yield with different type in Hunan Province in 2019.

**Author Contributions:** Conceptualization, X.Q.; methodology, X.Q.; software, Y.W.; validation, J.F.; formal analysis, Z.Z.; investigation, J.W., Y.L. (Yanhong Lu) and Y.L. (Yulin Liao); data curation, Z.Z.; writing—original draft preparation, Z.Z.; writing—review and editing, Z.Z. and X.Q.; supervision, Z.Z. and X.Q. All authors have read and agreed to the published version of the manuscript.

**Funding:** This study was supported by the National Key Research Plan (2021YFD1700202), Jiangxi Provincial Central Government Guide Local Science and Technology Development Fund Project (20221ZDH04057).

**Data Availability Statement:** The data are contained within the article.

**Conflicts of Interest:** The authors declare that they have no known competing financial interest or personal relationships that could appear to influence the work reported in this paper.

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
