# Peer review of "Evaluation of Methane Emission Reduction Potential of Water Management and Chinese Milk Vetch Planting in Hunan Paddy Rice Fields"

_agronomy, doi:10.3390/agronomy13071799_

Round 1

Reviewer 1 Report (New Reviewer)

This manuscript is interesting and of high importance. It will require additional editing to improve comprehension. A particular question/concern pertains to the Chinese Milk Vetch (CMV) field experiments. There is not sufficient detail to allow the reader to evaluate how these experiments were conducted. This is especially true for the methane analyses that were performed by GC. It is almost as if the CMV and associated methane analyses were reported elsewhere but that doesn't appear to be the case. Other suggested edits/comments are provided in the attached file. 

The manuscript would benefit from additional polishing of the English but is generally understandable. 

Author Response

Details for attachment please.

Reviewer 2 Report (Previous Reviewer 1)

Unfortunately, this paper does not contain relevant scientific data being based on local statistical data. It is unclear to me whether the main subject of this work is methane emission or potential methane emission from rice fields under different treatments, or the role of Chinese Milk Vetch used as a cover crop with or without straw under different water regimes on rice grain yield or on methane emission.

Also, the lack of statistical analysis disqualifies this paper as a scientific paper. Even if the data are very important and offer some preliminary positive answers, the structure and management of the experiment have major deficiencies.

Minor editing of English language required

Author Response

Dear reviewer,

Thank you for your suggestions. We believe that your suggestions will significantly improve paper quality. In the follow-up work, we will seriously consider your opinions and modify it.

Best regards,

Mr.Zhang

E-mail:[email protected]

Reviewer 3 Report (Previous Reviewer 2)

I have no more comments on the text, as it has been solidly completed.

Author Response

Dear reviewer,

Thank you for your suggestions. We believe that your suggestions will significantly improve paper quality. 

Best regards,

Mr.Zhang

E-mail:[email protected]

Round 2

Reviewer 2 Report (Previous Reviewer 1)

Unfortunately, the ambiguities specified in the previous review were not clarified. 

The English are ok, in my opinion.

Author Response

This manuscript is a resubmission of an earlier submission. The following is a list of the peer review reports and author responses from that submission.

Round 1

Reviewer 1 Report

Unfortunately, even if the subject is one of interest, the current work does not meet the rigors of a scientific paper. The lack of an experimental protocol and various uncontrollable factors load the experimental values with errors that cannot be quantified. So my opinion is that this work does not correspond to scientific work.

Reviewer 2 Report

General comments on methodology

The topic of the paper suggests a study of Chinese Milk Vetch, but in fact we know nothing about the Chinese Milk Vetch experiment, when, where, under what conditions it was conducted, in how many replicates, how methane emissions were determined in fields with Chinese Milk Vetch, whether it was harvested, whether it was left in the field and in what form (died in the field, mowed and left in the field or mowed and shredded, whether it was covered with soil, and if so, to what depth), whether its use affected yields. Only in line 225, without citing the source, is it indicated what the reductions in methane emissions were due to the use of this crop. Where did this data come from? Admittedly, it is indicated in the article that there was some sort of task group that "conducted a long-term field experiment on the effect of fertilizer and 194 emission reduction for super rice with the return of different organic materials under 195 equal nitrogen inputs in Changsha," i.e., they did not necessarily study fields with Chinese Milk Vetch. Nowhere in the paper about the effects of fertilization in the task group experiments is it mentioned what the yields were with different fertilization methods. Moreover, the fact that Changsha is in Hunan province we learn later.

There is some doubt about giving methane emissions for the entire Hunan province, because as the authors themselves point out in different regions emissions vary. Perhaps it would be better to give emissions per hectare, for example, but also per ton of crop, which is important in the context of the need to produce more and more food for a growing world population.

The summary after the addition of the methodological notes will only be consistent with the content of the

Other comments

Row 13 and 104 the purpose of the paper is not the same. Despite the large overlap, please provide one version of the study objective in these two places

Row 124 - yield should be changed to harvest, because from the whole area, not from a unit of area

Row 204-205 The details of the experiment are important, because they are crucial to the results obtained. Unfortunately, the source 18 cited by the authors is in Chinese? and the reviewer, and later the English-speaking reader, will not have knowledge in this regard. It would be appropriate, therefore, at least in a few sentences, to introduce this experiment

Row 221-223 277.66 Gg, accounting for 46.70% of the total emissions in the province. The 221 methane emissions in Huaihua, Loudi, Xiangtan and Xiangxi were relatively low, with a 222 total of 159.77 Gg, accounting for only 15.98% - are all values correct, because 15.98% AND 46.70% is a difference of more than 100%, and the differences in emissions are less than 100%. Similarly, rows 2209-231, except that no emissions in Gg for the 4 areas with the highest emissions.

Row 319. abbreviations used in the heading (CF; S; M; MS) should be explained below the table, because the reader may not be interested in reading the whole text or looking for the abbreviation designation in the text, and should know what is presented in the table.